# Feed Intake, Methane Emissions, Milk Production and Rumen Methanogen Populations of Grazing Dairy Cows Supplemented with Various C 18 Fatty Acid Sources

**DOI:** 10.3390/ani10122380

**Published:** 2020-12-11

**Authors:** Tommy M. Boland, Karina M. Pierce, Jason Rowntree, Alan K. Kelly, David A. Kenny, Mary B. Lynch, Sinéad M. Waters, Stephen J. Whelan, Zoe C. McKay

**Affiliations:** 1School of Agriculture and Food Science, University College Dublin, 4 D04 V1W8 Dublin, Ireland; karina.pierce@ucd.ie (K.M.P.); jason.rowntree@gmail.com (J.R.); alan.kelly@ucd.ie (A.K.K.); bridget.lynch@ucd.ie (M.B.L.); zoe.mckay@ucd.ie (Z.C.M.); 2Teagasc Animal and Bioscience Department, Animal & Grassland Research and Innovation Centre, Grange, Dunsany, C15 PW93 Co. Meath, Ireland; David.kenny@teagasc.ie (D.A.K.); sinead.waters@teagasc.ie (S.M.W.); 3Wexford Campus—IT Carlow, Summrhill, Y35 KA07 Wexford, Ireland; Stephen.Whelan@itcarlow.ie

**Keywords:** fatty acid, pasture, methane, milk composition, linseed oil, soy oil, stearic acid, C_18_ fatty acid

## Abstract

**Simple Summary:**

Reducing methane emissions from dairy cows is environmentally important. In this experiment, pasture fed dairy cows offered concentrates containing linseed oil emitted 18% less methane per kg of milk solids produced than those offered concentrates containing either stearic acid or soy oil. Additionally, cows fed linseed oil or soy oil produced more milk than those fed stearic acid. These results may contribute to the development of strategies to reduce methane emissions from pasture-based livestock whilst maintaining or improving animal productivity.

**Abstract:**

Emissions of methane (CH_4_) from dairy production systems are environmentally detrimental and represent an energy cost to the cow. This study evaluated the effect of varying C18 fatty acid sources on CH_4_ emissions, milk production and rumen methanogen populations in grazing lactating dairy cows. Forty-five Holstein Friesian cows were randomly allocated to one of three treatments (*n* = 15). Cows were offered 15 kg dry matter (DM)/d of grazed pasture plus supplementary concentrates (4 kg DM/d) containing either stearic acid (SA), linseed oil (LO), or soy oil (SO). Cows offered LO and SO had lower pasture DM intake (DMI) than those offered SA (11.3, 11.5 vs. 12.6 kg/d). Cows offered LO and SO had higher milk yield (21.0, 21.3 vs. 19.7 kg/d) and milk protein yield (0.74, 0.73 vs. 0.67 kg/d) than those offered SA. Emissions of CH_4_ (245 vs. 293, 289 g/d, 12.4 vs. 15.7, 14.8 g/kg of milk and 165 vs. 207, 195 g/kg of milk solids) were lower for cows offered LO than those offered SA or SO. *Methanobrevibacter ruminantium* abundance was reduced in cows offered LO compared to SA. Offering supplementary concentrates containing LO can reduce enteric CH_4_ emissions from pasture fed dairy cows.

## 1. Introduction

Estimates of the contribution of animal agriculture to total global anthropogenic greenhouse gas (GHG) emissions, ranges from 7% to 18% [1,2]. Emissions arising from enteric fermentation account for 19% of Ireland’s overall GHG emissions of which 58.9% is from agricultural emissions [3]. Irish agricultural derived GHG emissions increased by 2.9% in 2015 [4] reflecting the national strategy to increase milk production and the subsequent increase in dairy cow numbers [5]. However, despite the relatively high portion of GHG coming from ruminant agriculture, pasture-based systems are associated with lower GHG emissions compared with other farm systems [6]. In temperate regions of the world, pasture-based dairy production systems are associated with low feed production costs [7] due to the competitive cost advantage of including grazed pasture in the diet [8]. Indeed, increasing the proportion of grazed pasture in the diet of the dairy cow is a key component of sustainability in an EU post quota environment [9]. Furthermore, enhancing pasture quality reduces CH_4_ emissions from grazing systems [10,11,12], however, as the majority of pasture based dairy production systems in Ireland incorporate concentrate supplementation for at least part of the lactation [13], strategic manipulation of the chemical composition of supplemental concentrate offers further opportunity to reduce CH_4_ emissions. Potential to mitigate CH_4_ emissions via strategic lipid supplementation of grazed pasture diets [14] exists.

Lipid supplementation as a strategy to reduce enteric CH_4_ production is the subject of much study and review [15,16,17]. In particular, the addition of unsaturated fatty acids (UFA) such as soy oil (SO) or linseed oil (LO) to the diet have been reported to reduce both gross CH_4_ emissions measured as g/day (d) and on a per kg of production basis in beef [18] and dairy cattle [19,20]. In addition, dietary supplementation with UFA alters the rumen microbial community structure [21] with evidence suggesting that the alteration in community structure persists after withdrawal of the dietary supplement [22]. The bio-hydrogenation of UFA has been cited as a mechanism of poly unsaturated fatty acid (PUFA) derived CH_4_ mitigation, but stoichiometric analysis suggests that its contribution is small, with only 1–2% of the metabolic hydrogen available in the rumen being captured by this process [23,24]. Thus, the biological mechanism for the suppressive effect of UFA on ruminal methanogenesis is most likely due to the direct inhibitory effects of UFA on methanogenic Archaea [25,26,27] such as *Methanobrevibacter smithii*, *Methanobrevibacter ruminantium*, and *Methanosphaera stadtmanae* [28].

Meta-analyses by Moate et al. [29] and Grainger and Beauchemin [30] indicate a consistent decrease in ruminal CH_4_ production with dietary fat supplementation irrespective of its chemical nature [15]. The apparent greater potency of UFA compared with SFA on rumen microbial activity observed by Nagaraja et al. [31] has not been mirrored in terms of a reduction in CH_4_ production in most studies [32,33], although a greater mitigating effect of PUFA compared to SFA was observed in the analysis of Doreau et al. [34].

The hypothesis of this work is that the type of fatty acid (FA) included in the concentrate ration will alter rumen methane production, rumen microbial population and milk production and composition of pasture based lactating dairy cows. Therefore, the objectives of this study were (i) to determine the effects of PUFA vs. SFA supplementation on CH_4_ emissions and milk production in lactating dairy cows offered a pasture-based diet and (ii) to quantify the relative abundance of total methanogens and key species viz. *Methanobrevibacter smithii*, *Methanobrevibacter ruminantium*, *Methanosphaera stadtmanae,* and protozoa in the rumen fluid of cattle fed a diet supplemented with contrasting sources of dietary FA.

## 2. Materials and Methods

All procedures described in this experiment were approved by the animal research ethics committee at University College Dublin and conducted under experimental license from the Irish Department of Health and Children in accordance with the Cruelty to Animals Act 1876 and the European Communities (Amendment of the Cruelty to Animals Act 1876) Regulations 2002 and 2005 [35].

### 2.1. Experimental Design

Forty-five Holstein Friesian cows were selected from the spring calving dairy herd at University College Dublin Lyons Research Farm, Newcastle, Co. Dublin, Ireland (53°17′56″ N, 6°32′18″ W). Animals were blocked according to parity (24 multiparous and 21 primiparous), days in milk (DIM, 143 ± 22 d), and pre-experimental milk yield (24.6 ± 4.8 kg/d), and allocated to one of three treatments (*n* = 15) in a randomised block design. The treatments consisted of grazed pasture (15 kg DM/d allocated above 4 cm) plus 4 kg DM of a concentrate containing 16% fresh weight (FW) of either stearic acid (SA), SO or LO daily (Table 1; Gain Feeds, Glanbia PLC, Kilkenny, Ireland). Experimental concentrates were introduced to the cow gradually over a 7-d period (d -7 to 0) after which there was a 17-d dietary adaptation period prior to the first CH_4_ measurement period. Measurements were taken during two periods, where measurement period (MP) 1 corresponded to 17–22 d post treatment introduction and MP 2 corresponded to 44–49 d after treatment introduction. In total, the experiment ran for 64 d to allow for changes in body condition score (BCS) and body weight (BW) to be determined. The acclimatization period commenced on 09/07/2009 and the final BW measurement was conducted on 18/09/2009.

Cows were milked twice daily for the duration of the experiment at 0700 h and 1600 h. Recording and sampling of milk output during each MP was achieved using a milk meter system (Weighall; Dairymaster, Kerry, Ireland). Cow BCS on a scale of 1–5 (where 1 = emaciated and 5 = extremely fat), with increments of 0.25 was determined on d 0, 32, and 64 after treatment introduction according to Edmonson et al. [36] by a trained independent observer. Body weight was measured on two consecutive d on three occasions (d -1/0, d 31/32 and d 63/64) after morning milking using a calibrated electronic weighing scales (SR 3000; Tru-Test Ltd., Auckland, New Zealand). All cows had free access to water and were offered a new allocation (15 kg DM/d) of a perennial ryegrass (*Lolium perenne*) based sward once daily at 0800 h. Pre-grazing herbage mass was 1380 ± 310 kg DM/ha (available > 4 cm) and was determined weekly using the quadrant and shears method as described previously in Whelan et al. [37]. Supplementary concentrate treatments were offered using pre-programmed feed boxes (RIC System; Insentec B.V., Marknesse, The Netherlands) in equal allocations after milking which occurred at 0700 and 1600 h. The concentrates were formulated to be iso-lipogenic and to supply approximately 720 g/d of additional dietary lipid. The ingredients of the concentrate feeds offered and the chemical and FA composition of the offered herbage and concentrate feeds are presented in Table 1.

### 2.2. Feed Intake and Digestibility

Feed intake and apparent total tract digestibility were determined on a per cow basis during the two MP. Estimation of pasture dry matter intake (PDMI) was achieved using the n-alkane technique of Mays et al. [38] as described in Mulligan et al. [39]. Briefly, individual animals were dosed following am and pm milking with a paper bolus (Carl Roth, GmbH, KG, Karlesruhe, Germany) impregnated with 500 mg of the n-alkane n-dotriacontane (C_32_) for a period of 12 d. On the last 5 d of each MP, pasture herbage samples representative of that grazed by the cows were collected using a handheld shears (Gardena Accu 60; Gardena International GmbH, Ulm, Germany) during am and pm grazing. Pasture samples were then stored at −20 °C prior to analysis for chemical, FA and n-alkane analysis. When required for analysis, pasture samples were thawed at 4 °C for 16 h and bowl chopped (Hobart 842181, Hobart Corporation, Troy, OH, USA) to give a homogenous sample length. A portion of each sample was lyophilized (Labconco Freezone 18, Labconco Corp., Kansas City, MI, USA) for FA and n-alkane analysis with a further portion dried at 55 °C for 72 h in a forced air oven for DM determination. Fecal samples were collected during observation periods where cows naturally defecated and where not, samples were collected per rectum, using a lubricated gloved hand on the last 5 d following am and pm milking and dried at 55 °C for 72 h pending chemical analysis. Daily concentrate samples were collected, pooled on a per treatment basis for each MP and dried in a forced air oven at 55 °C for 72 h.

### 2.3. Enteric Methane Emissions

Total enteric CH_4_ emissions from the individual cows were determined daily during MP1 and MP2, using the emissions from ruminants using a calibrated tracer (ERUCT) technique, with sulphur hexafluoride (SF_6_) as the tracer gas as described by Johnson et al. [40]. Permeation tubes containing SF_6_ were blocked by SF_6_ emission rate (2.3 ± 0.3 mg/d) and randomly allocated to treatment and cow within treatment. The tubes were administered orally 7 d prior to MP1. Gas sampling commenced (0830 h) on the first d of each MP, where gas eructated by the cows, in addition to ambient air were drawn into an evacuated gas collection canister at a rate of 0.5 mL/min [41]. During each MP, canisters containing gas were removed from cows every 24 h (five times in total per MP) and replaced with pre-evacuated canisters. A pre-evacuated canister was placed on the fence, at each of the four corners of the area being grazed by the cows throughout each MP to determine background concentrations of CH_4_ and SF_6_. Concentrations of CH_4_ and SF_6_ in the gaseous contents of each canister were determined by gas chromatography (GC) using a Varian 3800 gas chromatograph (Varian Inc., Palo Alto, CA, USA) as described by Lovett et al. [42].

### 2.4. Milk Sampling and Analysis

During each MP, samples of milk were taken twice daily during consecutive p.m. and a.m. milking. Samples were pooled according to production on a per cow per week basis and concentrations of milk fat, protein and lactose were determined in a commercial milk laboratory (Progressive Genetics, Bluebell, Dublin 12, Ireland) using infrared analysis (CombiFoss 5000, Foss Analytical A/S, Hillerød, Denmark). Milk samples were prepared for analysis as per International Dairy Federation method 29-2:2004 [43].

### 2.5. Rumen Sample Collection and Analyses

On the final d of each MP, rumen fluid samples were harvested from each cow using a specialised trans-oesophageal sampler (FLORA Rumen Scup) within 1 h following concentrate feeding at both 0700 and 1600 h. Samples were strained through two layers of cheesecloth to remove feed particles, following which 20 mL was collected, snap frozen in liquid N_2_ and stored at −80 °C for microbial PCR analyses (Section 2.7). A further 8 mL was transferred into vials containing 2 mL of 50% trichloroacetic acid and stored at −20 °C prior to volatile fatty acid (VFA) and ammonia-nitrogen (NH_3_) analyses. Samples were prepared for analyses of VFA and NH_3_ as previously described by Whelan et al. [44]. The concentrations of individual VFA were determined using GC (Varian 3800; Varian Inc.) and NH_3_ using the phenol-hypochlorite method of Weatherburn [45].

### 2.6. Chemical Analysis of Feed and Feces

Dried pasture, feed and faeces samples were ground in a hammer mill fitted with a 1 mm screen (Lab Mill, Christy Turner, Suffolk, UK). The DM content of samples was determined by drying at 105 °C overnight for a minimum of 16 h [46]. Organic matter (OM) was calculated as 100—ash concentration; ash concentration was determined by complete combustion in a muffle furnace at 550 °C for 5 h [47]. Neutral detergent fiber (NDF) and acid detergnet fiber (ADF) were determined by the method of Van Soest et al. [48] using the Ankom 220 Fiber Analyzer (Ankom Technology, Fairport, NY, USA). Where the NDF content of concentrate feed was determined a thermo-stable α-amylase (17,400 Liquefon Units/mL, FAA, Ankom Technology, Fairport, NY, USA) was used. However, sodium sulphate was not used in the NDF procedure and both NDF and ADF are reported inclusive of residual ash. Acid detergent lignin (ADL) was determined following ADF determination by soaking the sample in 72% H_2_SO_4_ for 3 h and then triple rinsing with distilled H_2_O prior to drying at 104 °C for 3 h. The nitrogen (N) content of the pasture, concentrate and feces samples were determined by combustion (FP 528 Analyzer, Leco Corp, St Joseph, Michigan, USA) [46] and subsequently used to calculate crude protein (CP) concentration. Gross energy was determined by bomb calorimetry (Parr 1281 bomb calorimeter, Parr Instrument Company, Moline, Illinois, USA) whilst ether extract was determined using Soxtec instruments (Tecator, Höganäs, SE, Sweden) and light petroleum ether. The concentration of water soluble carbohydrates (WSC) was determined as described by Dubois et al. [49].

Alkane concentrations in faeces, freeze-dried pasture herbage, and concentrate feeds were determined using the extraction method of Dove and Mayes [50] with the modification that n-heptane was substituted for n-dodecane as the solvent used to rehydrate the extract prior to injection onto the gas chromatograph. Following extraction, samples were analyzed for concentration of n-alkanes by gas chromatography using a Varian 3800 GCL (Varian Inc.) fitted with a 30 m capillary column with an internal diameter of 0.53 mm coated with 0.5 μm dimethyl polysiloxane (SGE Analytical Science Pty LTD, Ringwood, Victoria, Australia). These data were then applied to the following formulae to calculate pasture DMI and DMD.
DMI = (Fi/Fj × [Dj + (Ic × Cj)] − [Ic × Ci])/(Hi − [Fi/Fj × Hj])(1)
DMD = 1 − (dietary C_35_ (mg/kg)/faecal C_35_ (mg/kg))(2)
where Fi and Fj are the concentrations of naturally occurring odd-chain (mainly feed derived C_33_) and even-chain (mainly dosed C_32_) n-alkanes in faeces respectively (mg/kg), Hi and Hj are the concentrations of natural odd-chain and even-chain n-alkanes in pasture respectively (mg/kg), Dj is the daily dose rate of even-chain n-alkanes (mg/kg), and Ic is the daily concentrate intake (kg/d) [39].

The FA in feed were extracted using the one-step methylation procedure of Sukhija and Palmquist [51] as described in Palladino et al. [52]. Concentrations of individual FA were determined using an adaptation of the method of Christie, (1982) [53] using a GC fitted with a 100 m capillary column with an internal diameter of 0.25 mm coated with 0.39 μm film (CP7420 Varian Inc.). The injector temperature was 250 °C, and the detector temperature was 300 °C. The initial oven temperature was 140 °C, which was maintained for 10 min. The oven temperature was then increased at a rate of 4 °C/min until it reached 240 °C, which was maintained for 17 min. The carrier gas was N_2_ and the column flow was maintained at 2 mL/min.

### 2.7. Relative Quantitative PCR Analysis of Ruminal Protozoa and Methanogenic Populations in Rumen Fluid

A detailed description of the DNA extraction methodology has been described by Carberry et al. [54]. Rumen microbial DNA was isolated from ruminal fluid using a repeated bead beating method outlined by Yu and Morrison [55]. The yield (ng μL^–1^) and purity of extracted DNA was assessed using a spectrophotometer (NanoDrop ND-1000; NanoDrop Technologies, Wilmington, DE). Specific primer and probe sets were used to detect and quantify protozoa [54] and dominant methanogen species *Methanobrevibacter ruminantium, Methanobrevibacter smithii, Methanosphaera stadtmanae*, and total methanogen populations (targeted using both rrs), along with a prokaryote rrs reference gene (Table 2) [28].

Quantitative real-time PCR was performed using either SYBR green chemistry (Fast SYBR green master mix; Applied Biosystems, Dublin, Ireland) or FAM dye (TaqMan; Life Technologies, Glasgow, UK) on the 7500 Fast Real-Time PCR System (Applied Biosystems). Real-time PCR amplification efficiencies (e) were estimated for all assays using a linear regression of the threshold cycle (Ct) for each dilution vs. the log dilution using the formula: e = 5–1/slope (Pfaffl, 2001) where “5” is the corresponding fold dilution.

Aliquots of 10 μL PCR products were analyzed by electrophoresis on a 2% agarose gel (wt/vol) to verify the presence and size of the amplicons. Negative controls without template DNA were included in parallel. The specificity of TaqMan assays for the quantification of *Methanobrevibacter smithii* and *Methanobrevibacter ruminantium* were verified before quantification of ruminal DNA. Each probe was validated by running a non-target clone standard as a negative control. Thermal cycling conditions applied to each assay consisted of an initial Taq activation step at 95 °C for 15 min, followed by 40 cycles of 95 °C for 15 s, 60 °C for 60 s, followed by an amplicon dissociation stage (95 °C for 15 s, 60 °C for 1 min, increasing 0.5 °C/cycle until 95 °C was reached), which confirmed specificity via dissociation curve analysis of PCR end products. Fluorescence detection was also performed at the end of each denaturation and extension step.

Inter-plate calibration, based on a calibrator sample included on all plates, efficiency correction of the raw cycle threshold (CT) values, and results from triplicate PCR reactions for each target species, were averaged and the means calculated, using the software package GenEx 5.2.1.3 (MultiD Analyses AB, Gothenburg, Sweden).

The abundance of protozoa and total methanogens were expressed as a proportion of total estimated ruminal bacterial 16S rDNA, as described previously [55,60], while specific methanogens were expressed as a proportion of total methonogens according to the equation: relative quantification = 2–(Ct target–Ct total bacteria or total methanogens), where Ct represents threshold cycle.

### 2.8. Statistical Analysis

Data were checked for normality and homogeneity of variance by histograms, qqplots, and formal statistical tests as part of the UNIVARIATE procedure of SAS. Data that were not normally distributed were transformed by raising the variable to the power of lambda. The appropriate lambda value was obtained by conducting a Box-Cox transformation analysis in the TRANSREG procedure of SAS. Values for NH_3_ concentration and iso-valeric acid concentration required a transformation and were raised to the power of 0.25. The transformed data were used to calculate *p*-values. However, the corresponding least squares means and SE of the non-transformed data are presented in the results for clarity. Data were analysed using the MIXED procedure of SAS. The statistical model used included the fixed effects of treatment (SA, SO, and LO), MP (1 and 2), parity (1 and >1) and the interaction between treatment and MP. The interaction term was subsequently excluded from the final model if not statistically significant (*p* > 0.05). Pre-experimental milk yield and DIM were included in the model as covariates. A random cow effect was included in the final model for all traits. Variables having more than one observation per subject were analyzed using repeated measures ANOVA (MIXED procedure), with terms for treatment, d, MP, parity and all possible interactions between treatment and d or MP. The interaction term was subsequently excluded from the final model if not statistically significant (*p* > 0.05). Pre-experimental milk yield and DIM were included in the model as covariates. The type of variance-covariance structure used was chosen depending on the magnitude of the Akaike information criterion (AIC) for models run under compound symmetry, unstructured, autoregressive, or Toeplitz variance-covariance structures. The model with the lowest AIC value was selected. Differences between treatments were determined by F-tests using Type III sums of squares. The PDIFF option and the Tukey test were applied as appropriate to evaluate pair wise comparisons between treatment means.

All rumen microbial data were analysed using Statistical Analysis Systems v9.1 2002 (SAS Institute, Cary, NC, USA). Data were examined for normality and homogeneity of variance by histograms, qqplots and formal statistical tests as part of the UNIVARIATE procedure of SAS. Data that were not normally distributed were transformed by raising the variable to the power of lambda. The appropriate lambda value was obtained by conducting a Box-Cox transformation analysis using the TRANSREG procedure in SAS. The relative abundance of total Methanogens was transformed using a lambda value of −0.5 while the relative Protozoa, *Methanosphaera stadtmanae, Methanobrevibacter smithii,* and *Methanobrevibacter ruminantium* abundances were transformed using a lambda value of 0.25. The transformed data were used to calculate *p*-values. However, the corresponding least squares means and standard errors of the non-transformed data are presented in the results for clarity. A mixed model ANOVA (PROC MIXED) was conducted to determine the effect of treatment on the relative abundance of each species of interest. In all analyses, the individual animal was denoted as the experimental unit, and animal within treatment was set as the error term.

## 3. Results

### 3.1. Feed Intake and Diet Digestibility

The effect of treatment on feed intake and digestibility parameters is presented in Table 3. Pasture DMI was higher (*p* = 0.02) for cows offered SA compared to those offered SO or LO which did not differ (*p* > 0.05). Total DM intake was greater for the SA group compared to the LO group (*p* = 0.01) with no other treatment differences recorded. Concentrate intake was not affected by treatment (*p* 0.05). Gross energy intake (GEI, MJ/d) was higher for cows offered SA compared to those offered SO or LO (*p* = 0.004) which were not different (*p* = 0.85). Measurement period had no effect on any of the feed intake related variables measured.

Dry matter, OM, CP, NDF or ADF did not differ (*p* > 0.05) between SO and LO supplemented animals (Table 3). The SA supplemented cows had lower DM (*p* = 0.02), and OM digestibility (*p* = 0.007) and higher CP (*p* = 0.001) digestibility than the LO group. Dry matter digestibility (*p* = 0.05) was higher and CP (*p* < 0.001) and NDF (*p* = 0.03) digestibility lower in MP1. Other digestibility parameters did not differ with MP.

### 3.2. Animal Performance

Milk yield and milk composition variables are presented in Table 4. Milk yield across the study was lower for the SA cows compared to SO and LO cows (*p* < 0.001). In addition, milk fat percentage was higher for the SA cows compared to SO and LO (*p* = 0.008) while milk fat yield (*p* = 0.60) was unaffected by treatment. Milk solids (fat and protein) yield was lower for SA compared to SO (*p* < 0.05). Milk protein percentage (*p* = 0.03) and milk protein yield (*p* < 0.001) was lower for SA compared to SO supplemented animals. Protein yield was higher for LO animals compared to SA animals (*p* < 0.001). There was no effect of treatment on BCS and BW change (*p* = 0.60 and *p* = 0.42, respectively).

### 3.3. Enteric Methane Emissions

The effect of treatment on CH_4_ emissions is presented in Table 5. Cows offered LO had lower daily CH_4_ emissions (g/d, *p* = 0.002), CH_4_ emissions per kg of milk (g CH_4_/kg milk; *p* < 0.001) CH_4_ emissions per kg of milk solids (g CH_4_/kg milk solids; *p* < 0.001), CH_4_ emissions per kg DMI (g CH_4_/kg DMI; *p* < 0.001) and CH_4_ emissions per g of added fat (g CH_4_/g of added fat; *p* = 0.02) than those offered SA or SO while cows offered SA and SO did not differ (*p* > 0.05). Methane emissions per unit of GEI (MJ CH_4_/MJ GEI) was lower for coes offered LO than SO (*p* = 0.002) with SA intermediate and not different (*p* > 0.05) from other treatments. All CH_4_ emissions variables were lower (*p* < 0.001) during MP1 than during MP2.

### 3.4. Rumen Fermentation Variables

The effect of fatty acid supplementation on rumen fermentation variables is presented in Table 6. Cows offered LO had lower concentrations of VFA in their rumen fluid compared to those offered SA (*p* = 0.04) with SO intermediate and not different from other treatments (*p* > 0.05). Butyric acid as a proportion of total VFA (*p* = 0.006) was lower for SO and LO relative to SA animals. Iso-valeric acid proportion was lower (*p* = 0.01) in the rumen fluid of cows on the SO treatment compared to LO with SA animals intermediate. The ratio of acetic acid to propionic acid did not differ with treatment (*p* = 0.14). Cows offered SA had a higher (*p* = 0.002) concentration of NH_3_ in rumen fluid compared to those offered LO while SO animals were intermediate (*p* = 0.47). Total VFA (*p* = 0.03) and NH_3_ (*p* < 0.0001) concentrations and proportion of iso-valeric acid (*p* = 0.01) in the rumen fluid were lower during MP1 than MP2. The proportion of butyric acid was lower in MP2 for the SA treatment (*p* = 0.04) while values for SO (*p* = 0.68) or LO (*p* = 0.99) did not differ due to period. There was no other treatment by MP interactions (*p* > 0.05) for any rumen fluid VFA or NH_3_ variable.

### 3.5. Ruminal Abundance of Protozoa and Methanogenic Populations

Table 7 shows the relative abundance of rumen microbial populations in cows with diets supplemented with LO compared to SA supplemented animals during MP1 and 2. There was no diet by period interaction for any microbial group examined (*p* = 0.169 to 0.60). In addition, there were no statistically significant differences observed in the relative abundance of total or specific methanogenic population over time (*p* = 0.18 to 0.67). However, the abundance of ruminal protozoa was approximately three-fold greater in cows in MP2 compared to MP1 (*p* = 0.03). Furthermore, the abundance of M. ruminantium was significantly (*p* = 0.01) reduced in the rumen of dairy cows offered the LO supplement compared to SA.

## 4. Discussion

Based on the observed data, we are confident that the stated hypotheses that FA type alters rumen CH_4_ production, rumen microbial population, milk production, and composition of pasture based lactating dairy cows can be accepted. Beauchemin et al. [61] reported that for each 1% addition of fat supplement, CH_4_ emissions per kg DMI were reduced by 5.6%. Diets in the current study were balanced for supplemental fat inclusion levels and were designed to supplement the pasture fed dairy cow with concentrates in which the FA profile was either predominantly SFA (SA) or UFA (SO and LO). The SO concentrate contained 53% C_18:2_ and 5% C_18:3_ in the FA profile whereas, LO contained 22% C_18:2_ and 42% C_18:3_ in the FA profile, thereby broadening the UFA composition of these treatments. From the results of the current study it is clear that FA type can have a major effect on enteric methanogenesis abatement as evidenced by the reductions reported when lactating dairy cows were supplemented with LO.

The important question of persistence of the effect of dietary lipid supplementation on ruminal CH_4_ production has not been adequately addressed in the literature. In a study with dairy cows on pasture, Woodward et al. [62] examined the effect of both vegetable and fish oils on milk production and CH_4_ emission after 14 d and again after 12 wk. Both lipid supplements significantly decreased CH_4_ production in the short term, but this effect was not observed after 11 weeks of supplementation. In the current study, a CH_4_ mitigation effect of LO supplementation was evident after a seven-week supplementation period. An additional CH_4_ measurement taken 28 days after the cessation of dietary oil supplementation in the current study (data not shown) indicated no difference in emissions between the SA and LO treatments highlighting that once the mitigating agent was removed from the diet there was no continued carry over effect on ruminal methanogenesis highlighting that the mitigating agent should be continuously supplemented.

### 4.1. Feed Intake and Diet Digestibility

Pasture DMI during this experiment (11.8 kg/d) was lower than that reported in other studies conducted at this institute; 15.74 kg/d at 92 DIM, 14.52 kg/d at 165 DIM and 16.83 kg/d at 220 DIM [63,64,65]. The cows used in this experiment were of a similar stage in lactation (181 DIM) and were offered similar levels of concentrate as those used in Mulligan et al. [39] thus ruling out concentrate substitution rate and DMI as sources of variation in PDMI. However, supplementing dairy cows with concentrate rich in FA has been shown to reduce the intake of forage [66] and may account for the lower PDMI observed in this study compared to earlier studies at this institute. As treatments in the current study were isolipogenic, it is not possible to confirm this finding in the current study.

Supplementary FA type also influences PDMI with lower intakes reported in the current study when the main fatty acid supplement was C18:0. Unsaturated FA are known to increase plasma concentrations of cholecystokinin and depress feed intake in dairy cattle offered high fat diets [67]. The higher PDMI of the SA group in the current study is consistent with observations of [68] where UFA were infused into the abomasum. It is likely that including high levels of UFA in the SO and LO treatments resulted in greater flow of UFA to the small intestine stimulating cholecystokinin production and a reduction in feed intake when compared to the SA diet.

Dry matter digestibility co-efficients were lower in the current study than previously reported from this institution where similar levels of concentrate supplementation at pasture were offered [44], highlighting the influence of fat inclusion on fibrolytic bacteria [31] and subsequently NDF and DM digestibility [69]. Nutrient digestibility (DM and NDF) was also altered by the type of FA supplement offered to the dairy cows. Degree of unsaturation of the FA and rate of release in the rumen are thought to be positively associated with a decrease in ruminal fermentation [70]. However, in the current study, LO supplemented animals demonstrated a greater ability to degrade DM and NDF than animals supplemented with SA. This agrees with Beauchemin et al. [32] who reported that UFA were not more harmful to fibre digestion.

Dietary unsaturated C_18_ FA are inhibitory to rumen protozoa and can greatly reduce rumen protozoa numbers [71,72,73,74], thus altering rumen protein metabolism [75]. In the current study, higher CP digestibility in SA fed animals is consistent with an increase in rumen NH_3_ concentration and suggests reduced rumen protein metabolism in SO and LO fed animals compared to those offered SA.

### 4.2. Enteric Methane Emissions

There is a strong relationship between total DMI, the concentration of VFA in the rumen fluid [75] and total CH_4_ emissions [76]. Therefore, it is not surprising that the cows offered LO had lower tCH_4_ than those offered SA as DMI and concentrations of VFA in the rumen fluid were higher in the latter treatment. However, tCH_4_ emissions were also lower for LO fed cows compared to SO fed cows despite similar levels of DMI and total VFA concentrations in the rumen fluid between the two treatments. Beauchemin et al. [77] also reported that linseed oil had a greater mitigating effect on CH_4_ emissions by dairy cows than a C_18:2_ rich supplement (sunflower seeds). In the current study a similar finding occurred where the C_18:3_ rich linseed oil, had a greater mitigating effect than the C18:2 supplement (soy oil). Similarly, there was no effect of treatment on the individual portions of VFA in the rumen fluid, reducing the potential for propionic acid to act as an alternate H_2_ sink to methanogenesis by rumen archaea [78,79]. Unsaturated FA are also potential sinks for H_2_ due to bacterial biohydrogenation of these FA. The LO treatment in this experiment contained higher concentrations of C_18:3_ (42% total FA) compared to the SO (5% total FA). As C_18:3_ has a greater degree of unsaturation than C_18:1_ and C_18:2_ [80] it most likely underwent more extensive rumen biohydrogenation as biohydrogenation increases with increasing degree of C_18_ FA unsaturation [81,82]. This in turn may have resulted in a lower amount of H_2_ available to methanogenic Archaea resulting in the lower tCH_4_ for LO fed animals. The effect of degree of unsaturation of FA on CH_4_ emissions is further emphasised when CH_4_ emissions are examined on a per kg of FA added to the diet with animals offered LO having lower CH_4_/kg FA supplementation compared to those offered other treatments. This is an important finding as FA supplementation is expensive and identifying oils with greater CH_4_ reducing potential may allow for a reduction in the total amount of FA added to the diet.

When reporting emissions of livestock origin, it is common to express these emissions on a unit of production or emissions intensity basis. As the animals offered LO in this study had lower total CH_4_ emissions and higher milk yield than those offered SA, CH_4_/kg milk was also lower regardless of MP. However, MP did influence total CH_4_ emissions with MP2 being greater than MP1. This may have been partially due to an increase in NDF digestibility resulting in greater H_2_ availability for methanogenesis consistent with the numerically higher VFA concentrations observed for MP2 when compared to MP1. It is also possible that the elevated CH_4_ emissions in MP2 compared to MP1 are because of inherent variation in the SF_6_ technique [83,84,85]. At the time the study was conducted, the SF_6_ methodology applied had been widely validated by several users [86,87]. However, Deighton et al. [84] subsequently showed that the pre-calibrated SF_6_ release rate declines over time rendering the assumption of linearity of SF_6_ release redundant. This would have contributed to an overestimation of CH_4_ emissions as witnessed in the increase total CH_4_ values from MP1 to MP2. However, as SF_6_ boluses were blocked across treatments according to release rate, in the current study, this would not have influenced the interpretation of the effect of FA supplementation on ruminal CH_4_ emissions.

### 4.3. Animal Performance

Supplements which affect nutrient digestion and rumen fermentation are also likely to affect milk production through altering the quantity and type of substrate available for de novo synthesis. Additionally, feeding UFA to ruminant animals results in the production of specific FA that are formed in the rumen as intermediates in the biohydrogenation of UFA by the rumen microbes [88]. Of these FA, trans-10, cis-12 CLA been shown to induce milk fat depression, possibly through decreased expression of genes that encode for enzymes involved in FA uptake and transport, de novo FA synthesis, desaturation of FA, and triglyceride synthesis [89,90]. The combination of a lower DMI and higher UFA intake in SO and LO fed animals compared to SA fed animals likely reduced substrate supply for de novo synthesis of milk fat creating a difference in the milk fat content of animals offered these diets versus the SA fed animals. However, milk fat yield was not affected by dietary treatment as both SO and LO fed animals gave higher milk yields. This is important from a dairy producer’s perspective as milk is often valued based on constituent yield. Indeed, the improvement in milk yield in SO and LO fed animals despite these animals having lower DMI indicates that the energy consumed by these diets was more efficiently used for productive purposes, consistent with the reduction in total CH_4_ emitted from these animals and earlier reports [91].

Improvements in the N efficiency in the rumen are also likely to have positive effects on milk production, and in particular milk protein production [92]. Indeed, many authors have examined the effect of optimizing the ratio of fermentable carbohydrates to fermentable N in the rumen as a means of improving the N efficiency within the organ [93,94]. However, in this study, the addition of UFA to the diet of the dairy cow appeared to manipulate the N ecology of the rumen as NH_3_ concentrations were lower for LO and SO fed animals most likely due to the direct inhibitory effects of UFA on rumen protozoa [71,73]. It is likely that this inhibition resulted in greater microbial N flow to the small intestine [71] thus improving the supply of amino acids for milk protein synthesis consistent with the improvement in milk protein for SO and LO fed animals. Despite differences in GE intake among treatments, both BCS and body weight were un-affected and it is most likely that the scale of these differences in energy partitioning were too small to detect as changes in either BCS or body weight.

### 4.4. Rumen Methanogen Populations

The relative abundance of *M. ruminantium* was significantly reduced by 50% following supplementation with linseed oil. Total and other specific methanogens were numerically reduced following supplementation but due to large variation in species abundance between individual animals, this reduction did not reach statistical significance. Similarly, Li et al. [95] reported a 49% reduction of archaea specific 16S rRNA and 50% reduction of the functional mcrA gene abundances in the rumen of dairy cows fed the linseed oil supplemented diet in comparison with those fed the control diet, demonstrating that supplementation effectively decreases the methanogenic population in the rumen. However, this group did not focus on the specific methanogenic species as carried out in the current study.

There was an increase in the relative abundance of protozoa over time. Furthermore, ruminal protozoal abundance had a positive relationship with rumen butyric acid. This is consistent with findings of Morgavi et al. [96] who found that the absence of protozoa in defaunated sheep resulted in a reduction in ruminal concentration of butyrate. Morgavi et al. [97] showed a positive relationship between methane emissions and protozoal numbers as also demonstrated in current study. Similar to other investigations [56], we did not observe a relationship between methane emissions and relative abundance of total methanogen. However, there was a positive relationship between abundance of both total methanogens and protozoa. This positive relationship is compounded by the fact that among H_2_ producers, protoza have a prominent position, which is strengthened by their close physical association with methanogens, which favors H_2_ transfer from one to the other [97].

There were strong positive relationships between the abundance of total and specific methanogens which was reassuring. Similarly, we recently showed a strong positive relationship between total methanogens and *M. smithii* [28].

## 5. Conclusions

Supplementing pasture fed dairy cows with concentrate containing LO reduces total CH_4_ emissions and CH_4_ per unit of milk produced compared with SO and SA. Increasing the UFA intake resulted in greater milk yield of higher protein and lower fat concentrations. Lastly, we have demonstrated that the degree of unsaturation of FA is an important consideration when choosing dietary strategies to reduce CH_4_ emissions—those with greater unsaturation having the biggest impact on the extent of ruminal methanogenesis and the abundance of *Methanobrevibacter ruminantium*. Offering concentrate supplementation containing linseed oil in particular has the potential to be used as an effective CH_4_ mitigation strategy, without negatively affecting the performance of dairy cows in pasture based dairy production systems.

## Figures and Tables

**Table 1 animals-10-02380-t001:** Concentrate ingredients and chemical and fatty acid composition of the concentrates and pasture offered.

Items	Pasture Herbage		Concentrate Feed ^2^		
	MP ^1^ 1	MP 2	SA	SO	LO
Ingredients (%, as fed)	-	-	-	-	-
Barley	-	-	28.5	28.5	28.5
Citrus pulp	-	-	25.0	25.0	25.0
Soy bean meal	-	-	20.4	20.4	20.4
Stearic acid	-	-	16.0	-	-
Linseed oil	-	-	-	-	16.0
Soy oil	-	-	-	16.0	-
Cane molasses	-	-	6.0	6.0	6.0
Minerals and vitamins	-	-	4.1	4.1	4.1
Chemical composition (% of DM ^3^, unless stated)					
DM (% as fed)	22.2	17.2	88.6	88.5	89.6
OM	92.3	91.4	91.9	91.5	92.7
CP	11.5	16.7	16.6	16.4	15.9
NDF	50.2	55.0	22.9	22.6	22.5
ADF	27.0	28.7	10.1	11.7	10.5
ADL	2.3	2.5	1.6	2.7	2.5
WSC (% as fed)	4.7	2.5	-	-	-
Ether extract	2.6	3.1	1.4	1.3	1.8
Gross energy (MJ/kg DM)	18.4	18.7	19.9	18.6	20.3
Fatty Acids (% of total fatty acids)					
Tetradecanoic (C_14:0_)	1.8	2.0	0.2	0.3	0.1
Hexadecanoic (C_16:0_)	19.2	18.3	6.5	12.5	7.5
c/t-9-Hexadecanoic (C_16:1_)	1.8	2.4	0	0	0
Octadecanoic (C_18:0_)	0.5	0.5	80.4	2.8	3.7
c/t-9-Octadecanoic (C_18:1_)	1.2	1.0	1.6	23.5	22.1
c-9,12-Octadecanoic (C_18:2_)	9.7	10.7	6.9	52.9	22.1
Octadecatrienoic (C_18:3_)	34.8	44.1	2.7	5.2	42.4
Others	31.0	21.0	1.7	2.8	2.1

^1^ MP = measurement period, MP 1 corresponds to 17 to 22 d post treatment introduction and MP 2 corresponds to 44 to 49 d after treatment introduction; ^2^ SA = supplementary concentrate containing 16% FW stearic acid; SO = supplementary concentrate containing 16% FW soy oil; LO = supplementary concentrate containing 16% FW linseed oil; ^3^ DM = dry matter; OM = organic matter; CP = crude protein; NDF = neutral detergent fiber; ADF = acid detergent fiber; ADL = acid detergent lignin; WSC = water soluble carbohydrates.

**Table 2 animals-10-02380-t002:** PCR primers and probes used for the quantification of ruminal protozoal and methanogenic microbial populations by qPCR analysis.

Target	Primer/Probe Name and Sequence	Assay	Efficiency (%)	Reference
Total bacteria (rrs)	V3-16S-F, 5′-CCTACGGGAGGCAGCAG-3′ V3-16S-R, 5′-ATTACCGCGGCTGCTGG-3′	SYBR	96	[56]
Total methanogens (rrs)	Met630F, 5′-GGATTAGATACCCSGGTAGT-3′	SYBR	92	[57]
	Met803R, 5′-GTTGARTCCAATTAAACCGCA-3′			
Total prokaryotes (rrs; reference gene)	V3-F, 5′-CCTACGGGAGGCAGCAG-3′	SYBR	91	[56]
	V3-R, 5′-ATTACCGCGGCTGCTGG-3′			
*M. stadtmanae* (rrs)	Stad-F, 5′-CTTAACTATAAGAATTGCTGG-3′	SYBR	98	[58]
	Stad-R, 5′-TTCGTTACTCACCGTCAAGAT-3′			
*M. smithii* (rrs)	Smit.16S-740F, 5′-CCGGGTATCTAATCCGGTTC-3′	FAM ^1^	91	[59]
	Smit.16S-862R, 5′-TCCCAGGGTAGAGGTGAAA-3′			
	Smit.16S FAM, 5′CGTCAGAATCGTTCCAGTCA-3′			
*M. ruminantium* (rrs)	Rum16S 740F, 5′-TCCCAGGGTAGAGGTGAAA-3′	FAM	92	[28]
	Rum16S 862R, 5′CGTCAGAATCGTTCCAGTCA-3′			
	Rum16S FAM, 5′-CCGTCAGGTTCGTTCCAGTTAG-3′			
Protozoa	Prot 18S F 5′-GCTTTCGWTGGTAGTGTATT-3′	SYBR	95	[54]
	Prot 18S R 5′-CTTGCCCTCYAATCGTWCT-3′			

^1^ SYBR = Fast SYBR Green I Dye assay; FAM = TaqMan probe-based assay.

**Table 3 animals-10-02380-t003:** Effect of C_18_ fatty acid source in the supplementary concentrate and measurement period on feed intake and nutrient digestibility of lactating dairy cows offered a pasture based diet (least square means ± SEM).

Items	Treatment ^1^				Measurement Period (MP) ^2^				*p*-Value	
	SA	SO	LO	SEM		1	2	SEM	Treatment	MP
Intake (kg DM ^3^/d)										
Pasture	12.6 ^a^	11.5 ^b^	11.3 ^b^	0.32		11.9	11.6	0.26	0.02	0.53
Concentrate	4.0	4.1	3.9	0.08		4.0	4.0	0.06	0.09	0.63
Total	16.6 ^a^	15.6 ^a,b^	15.2 ^b^	0.31		15.9	15.6	0.26	0.01	0.44
GEI ^3^ (MJ/d)	315 ^a^	292 ^b^	287 ^b^	5.8		298	298	4.7	0.004	0.90
Digestibility (%)										
DM ^3^	66.9 ^b^	67.6 ^a,b^	69.1 ^a^	0.05		68.5	67.3	0.04	0.02	0.05
OM ^3^	70.1 ^b^	71.0 ^a,b^	72.4 ^a^	0.05		71.4	70.9	0.04	0.007	0.45
CP ^3^	64.6 ^a^	61.2 ^b^	62.3 ^b^	0.06		58.9	66.5	0.05	0.001	<0.001
NDF ^3^	63.4 ^a^	64.7 ^a,b^	66.3 ^b^	0.09		63.7 ^b^	65.9	0.07	0.07	0.03
ADF ^3^	62.0	62.5	64.8	0.01		62.7	63.5	0.08	0.12	0.46

^1^ SA = supplementary concentrate containing 16% FW stearic acid; SO = supplementary concentrate containing 16% FW soy oil; LO = supplementary concentrate containing 16% FW linseed oil; ^2^ MP = measurement period, MP 1 corresponds to 17 to 22 d post treatment introduction and MP 2 corresponds to 44 to 49 d after treatment introduction; ^3^ DM = dry matter; OM = organic matter; CP = crude protein; NDF = neutral detergent fiber; ADF = acid detergent fiber; ADL = acid detergent lignin; WSC = water soluble carbohydrates. ^a,b^ Means within rows differ significantly.

**Table 4 animals-10-02380-t004:** Effect of C_18_ fatty acid source in the supplementary concentrate on milk production and milk composition body weight and body condition score of lactating dairy cows offered a pasture based diet (least square means ± SEM).

Items	Treatment ^1^				
	SA	SO	LO	SEM	*p*-Value
Milk yield (kg/d)	19.7 ^b^	21.3 ^a^	21.0 ^a^	0.21	<0.001
Fat yield (kg/d)	0.85	0.84	0.83	0.014	0.60
Protein yield (kg/d)	0.67 ^b^	0.73 ^a^	0.74 ^a^	0.008	<0.001
Fat + Protein yield ^2^ (kg/d)	1.50 ^a^	1.58 ^b^	1.56 ^a,b^	0.018	<0.05
Fat %	4.14 ^a^	3.89 ^b^	3.88 ^b^	0.068	0.008
Protein %	3.29 ^a^	3.42 ^b^	3.36 ^a,b^	0.032	0.03
BCS ^3^	2.8	2.9	2.8	0.05	0.60
BW ^4^ change (kg/d)	0.16	0.16	0.05	0.09	0.42

^1^ SA = supplementary concentrate containing 16% stearic acid; SO = supplementary concentrate containing 16% soy oil; LO = supplementary concentrate containing 16% linseed oil; ^2^ Fat and protein yield (kg) = the sum of; ^3^ BCS = body condition score on a scale of 1 to 5 in 0.25 increments [37]; ^4^ BW = body weight. ^a,b^ Means within rows differ significantly.

**Table 5 animals-10-02380-t005:** Effect of C_18_ fatty acid source in the supplementary concentrate and measurement period on methane emissions from lactating dairy cows offered a pasture based diet (least square means ± SEM).

Methane (CH_4_) Variable	Treatment ^1^				Measurement Period (MP) ^2^			*p*-Value		
	SA	SO	LO	SEM	1	2	SEM	Treatment	MP	Treatment x MP
Total CH_4_, g/d	293 ^a^	289 ^a^	245 ^b^	8.9	241	311	7.2	0.002	<0.001	0.15
g CH_4_/kg milk	15.7 ^a^	14.8 ^a^	12.4 ^b^	0.5	12.3	16.3	0.4	<0.001	<0.001	0.14
g CH_4_/kg milk solids	207 ^a^	195 ^a^	165 ^b^	6.3	166	211	5	<0.001	<0.001	0.12
g CH_4_/kg DMI ^3^	17.9 ^a^	18.7 ^a^	16.3 ^b^	0.51	15.2	20.0	0.42	0.004	<0.001	0.02
g CH_4_/g of added FA ^4^	0.46 ^a^	0.45 ^a^	0.40 ^b^	0.16	0.38	0.49	0.01	0.02	<0.001	0.19
CH_4_/GEI ^5^, MJ/MJ	0.052 ^a,b^	0.055 ^a^	0.047 ^b^	0.0015	0.045	0.058	0.0012	0.002	<0.001	0.02

^1^ SA = supplementary concentrate containing 16% FW stearic acid; SO = supplementary concentrate containing 16% FW soy oil; LO = supplementary concentrate containing 16% FW linseed oil; ^2^ MP = measurement period, MP 1 corresponds to 17 to 22 d post treatment introduction and MP 2 corresponds to 44 to 49 d after treatment introduction; ^3^ DM = dry matter; OM = organic matter; CP = crude protein; NDF = neutral detergent fiber; ADF = acid detergent fiber; ADL = acid detergent lignin; WSC = water soluble carbohydrates; ^3^ DMI = dry matter intake; ^4^ FA = fatty acids; ^5^ GEI = gross energy intake. ^a,b^ Means within rows differ significantly.

**Table 6 animals-10-02380-t006:** Effect of C_18_ fatty acid source in the supplementary concentrate and measurement period on rumen fermentation parameters of lactating dairy cows offered a pasture based diet (least square means ± SEM).

Items	Treatment ^1^				Measurement Period ^2^			*p*-Value		
	SA	SO	LO	SEM	1	2	SEM	Treatment	MP	Treatment x MP
Total VFA ^3^, mmol/L	104 ^a^	97 ^a,b^	93 ^b^	3.1	94	102	2.5	0.04	0.03	0.65
Individual VFA proportions in total VFA										
Acetic acid	0.67	0.68	0.67	0.003	0.67	0.67	0.003	0.14	0.62	0.43
Propionic acid	0.18	0.17	0.18	0.003	0.18	0.17	0.002	0.13	0.21	0.93
Iso-butyric acid	0.0071	0.0069	0.0078	0.00033	0.0066	0.0081	0.00027	0.11	<0.001	0.58
Butyric acid	0.1300 ^a^	0.1242 ^b^	0.1215 ^b^	0.00185	0.1254	0.1251	0.00150	0.006	0.68	0.04
Iso-valeric acid	0.0086 ^a,b^	0.0077 ^a^	0.0093 ^b^	0.00036	0.0080	0.0091	0.00030	0.01	0.01	0.98
Valeric acid	0.0092	0.0095	0.0097	0.00022	0.095	0.0094	0.00021	0. 40	0.81	0.88
Acetic:propionic	3.81	3.98	3.77	0.079	3.82	3.89	0.065	0.14	0.49	0.67
NH_3_ ^4^, mg/L	55.0 ^a^	40.1 ^a,b^	32.6 ^b^	4.41	25.	60.0	3.48	0.002	<0.001	0.12

^1^ SA = supplementary concentrate containing 16% FW stearic acid; SO = supplementary concentrate containing 16% FW soy oil; LO = supplementary concentrate containing 16% FW linseed oil; ^2^ MP = measurement period, MP 1 corresponds to 17 to 22 d post treatment introduction and MP 2 corresponds to 44 to 49 d after treatment introduction; ^3^ VFA = volatile fatty acids; ^4^ NH_3_ = ammonia. ^a,b^ Means within rows differ significantly.

**Table 7 animals-10-02380-t007:** Effect of linseed oil supplementation on ruminal Protozoa and Methanogen populations ^1^.

Items	Treatment ^2^				Measurement Period ^3^			*p*-Value	
	SA	LO	SEM	MP1	MP2	SEM	Treatment	MP	Treatment x MP
Methanogens	1.9877	1.7332	0.2744	2.1733	1.5477	0.2905	0.311	0.18	0.215
Protozoa	6.3317	6.7198	1.2730	3.2446	9.8068	1.1562	0.806	0.028	0.601
*Methanosphaera stadtmanae* ^4^	0.0184	0.0061	0.00027	0.0100	0.00560	0.00029	0.19	0.21	0.169
*Methanobrevibacter smithii* ^4^	0.2248	0.1169	0.00439	0.1824	0.1497	0.0062	0.12	0.67	0.18
*Methanobrevibacter ruminantium* ^4^	0.1796	0.0879	0.00371	0.1776	0.0920	0.0038	0.011	0.326	0.269

^1^ Methanogens and Protozoa measured as relative abundance = 2^−(Ct target-Ct total bacteria)^ × 10^2^; 2 SA = supplementary concentrate containing 16% FW stearic acid; LO = supplementary concentrate containing 16% FW linseed oil; ^3^ MP = measurement period, MP 1 corresponds to 17 to 22 d post treatment introduction and MP 2 corresponds to 44 to 49 d after treatment introduction; ^4^ Methanogen spp. Measured as a proportion of total estimated rumen methanogen 16S rDNA, relative quantification = 2^−(Ct target-Ct total methanogens)^.

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
