# Peer review of "Feed Intake, Methane Emissions, Milk Production and Rumen Methanogen Populations of Grazing Dairy Cows Supplemented with Various C 18 Fatty Acid Sources"

_animals, 2020, doi:10.3390/ani10122380_

Round 1

Reviewer 1 Report

I think this is a very good paper overall. In my opinion, there are only minor things that can be improved.

line 28: Although this may be accepted, I'd suggest avoiding abbreviations within brackets. Consider something like: "Cows were offered 15 kg dry matter (DM)/ d of grazed pasture plus...". This also applies to line 62 "(g/day (d))".

line 32: values presented between brackets are unclear (value1 vs value2, but there are 3 treatments). Please specify clearly to which treatment every mean relate to

line 82 (and throughout): fatty acid was already abbreviated with FA. Please double check use of abbreviation "FA" throughout the manuscript

line 99-100: please specify whether pasture allocation is defined as total (above ground) or available (>3.5/4cm)

line 105-106 (ore somwehre else in this section): please report start and end dates of the experiment.

line 108-109: move model of milk meter system within the brackets.. Like "...using a milk meter system (Weighall; Dairymaster, Kerry, Ireland)."

line 109-110: please, along with the reference, provide a short description of the BCS scale used. You might report description of the lowest and highest value between brakets

line 114-115: please report frequency of Pre-grazing herbage mass determination (weekly?)

line 116-118: was the concentrate offered before or after milking? If I'm not wrong, the RIC system cannot be installed in the parlour...

line 289-290: "...as described in the previous statistical section." is not completely clear as there is only one section for statistical analysis

line 350 (and throughout): here the P (value) is upper case and italicized while is lowercase and italicized in the rest of the section (and manuscript). Please select one form and use it consistently. Please double check this also in the tables.

line 362: italics for "... M. ruminantium..."

line 473-481: both forms "de-novo" and "de novo" (italicized) are used. Please select one form and use it consistently.

line 518-519: words protozoa and methanogens are highlighted (probably from internal revision, or there's another reason?)

line 524: lineseed oil was already abbreviated as LO (again, please double check use of abbreviations)

line 524-526: I think in the conclusions it might be helpful to report other feedstuffs/treatments used in the experiment. So something like: "...LO reduces total CH4 emissions and CH4 per unit of milk produced as compared with SO and SA..."

Author Response

Dear Reviewer 1,

Thank you for your considered and detailed review of our manuscript. Please see the attachment for a detailed response to your queries. I hope everything is adequately addressed.

Sincerely

Prof. Tommy Boland 

Reviewer 2 Report

Article presents a very interesting and complete study about the effects of lipid supplementation on cow CH4 emissions, performance and rumen microbiology. 

Please, find below some comments/suggestions to address before considering the manuscript for publication:

L17 There is no need to include a short title and author’s name in the simple summary

L31 here and throughout the paper, please italicize “vs.

L72 references 29 and 30 are missing here, please, recheck reference numbering

L78 please, change (37) to [37]

L81 change “to (i)” to “(i) to”

L106 please, define BCS and BW

L114 Please, italicize “Lolium perenne

L173 change “N2” to “N2

L223-224 Please, italicize species names

L234-235 Please, italicize species names

L247 Please, include the sequence of the primer used to quantify total bacteria in the table

L259 change NH3 to NH3

L282 Please, specify that this is “total Methanogens relative abundance”. Same for the rest

L297 According to your table p=0.02.

Maybe it is worthy to mention that concentrate intake was not affected by treatment

L302 Please, define OM, CP, NDF, ADF

L303 Again, according to your table p=0.001

L305 p=0.03 for the NDF

Table 3: there is no need of superscripts in the values of CP digestibility of the measurement periods

L329-335 Table 4. Please, check carefully; the values presented do not correspond to the parameters listed in the Table (you copied Table 3 and did not change the values…)

L325-340 please, rewrite the paragraph. Results could be presented in an easier way. According to your table LO cow had lower values of all parameters than SA or SO cows with no difference between SA and SO, except for CH4/GEI

L342 Please change “are” to “is”

L346 Please, avoid using “reduced” as you don’t know for sure if it was reduced in this groups or increased in the other

L348 Please change “NH3” to NH3

L351-352 so there was a significative interaction, why didn’t you present the interaction P-values in the Table?

L353 Please, change “NH3” to “NH3

L355-356 You compared only SA and LO but, according to Table 5, LO cows had also lower emissions than SO cows, so I don’t see the reason to exclude them from the analysis. I suggest to include also values from SO treatment in the results section or to explain better why these samples were excluded from the microbial analysis

L359 Please, check p-values, the lowest in the Table is 0.169.

L362 Please, italicize M. ruminantium

L365 Table 5: Please, be consistent in the use of “Trt” or “Treatment” in the tables

L380 I would avoid using the term “proportion” as what you are expressing is not the proportion of these microorganisms. Please, use “abundance” or “relative abundance”. Also, specify that specific methanogens were determined relative to total methanogens.

L406 what is CO? If you mean SO, what about the SA group? Was it different from LO?

L411 Pease, define DIM

L418 Please, change “currently” to “current”

L444 According to your table VFA were higher in SO than in LO. Please, check if superscripts are correct

L451-453 VFA were lower for the LO group, so another possible explanation for the decrease in CH4 emissions is a slight inhibition of ruminal fermentation?

L462 Please change “H2” to “H2

L500-522 I would like to check this section when data on relative abundance of the SO group was also included in the results. Also, you state in the M&M that you perform correlations but no results are presented. Then, here in the discussion you discuss about certain “relationships” between protozoa and methanogens or total and specific methanogens, are those significant correlations?

L512 this is the first time this reference [29] appears in the text, please, recheck reference numbering. Same for reference 30.

L518-519 the are two words highlighted

L525 I would add “compared to cows supplemented stearic acid or soy oil”

L546 the recommended citation for this reference is:

Hristov, A.N., Oh, J., Lee, C., Meinen, R., Montes, F., Ott, T., Firkins, J., Rotz, A., Dell, C., Adesogan, A., Yang, W., Tricarico, J., Kebreab, E., Waghorn, G., Dijkstra, J. & Oosting, S. 2013. Mitigation of greenhouse gas emissions in livestock production – A review of technical options for non-CO2 emissions. Edited by Pierre J. Gerber, Benjamin Henderson and Harinder P.S. Makkar. FAO Animal Production and Health Paper No. 177. FAO, Rome, Italy.

Author Response

Dear Reviewer 2,

Thank you for your considered and detailed review of our manuscript. Please see the attachment for a detailed response to your queries. I hope everything is adequately addressed.

Sincerely

Prof. Tommy Boland 

Round 2

Reviewer 2 Report

Authors have correctly addressed all the remarks from my report, and I can recommend the paper for publication. I've only detected a spelling mistake, line 327 "coes" instead of "cows".

Author Response

Thank you for your positive reply and we have made the edit suggested